# The Pattern of Hemoglobin A1C Trajectories and Risk of Herpes Zoster Infection: A Follow-Up Study

**DOI:** 10.3390/ijerph19052646

**Published:** 2022-02-24

**Authors:** Bo-Lin Pan, Chia-Pei Chou, Kun-Siang Huang, Pin-Jie Bin, Kuei-Hau Luo, Hung-Yi Chuang

**Affiliations:** 1Department of Family Medicine, Kaohsiung Chang Gung Memorial Hospital, Kaohsiung 83301, Taiwan; samohtte@cgmh.org.tw (B.-L.P.); libra760924@cgmh.org.tw (C.-P.C.); b9402019@cgmh.org.tw (K.-S.H.); u107575105@kmu.edu.tw (P.-J.B.); 2Graduate Institute of Medicine, College of Medicine, Kaohsiung Medical University, Kaohsiung 80708, Taiwan; u107800007@kmu.edu.tw; 3Department of Community Medicine, Kaohsiung Medical University Hospital, Kaohsiung 80756, Taiwan; 4Ph.D. Program in Environmental and Occupational Medicine, Research Center for Environmental Medicine, Department of Public Health and Environmental Medicine, College of Medicine, Kaohsiung Medical University, Kaohsiung 80708, Taiwan

**Keywords:** type 2 diabetes mellitus, herpes zoster, glycated hemoglobin, group-based trajectory model (GBTM)

## Abstract

To investigate the risks of herpes zoster (HZ) infection among heterogeneous HbA1C trajectories of patients with newly diagnosed type 2 diabetes, this cohort study used data from the Chang Gung Research Database (CGRD), from the 10-year period of 1 January 2007 to 31 December 2017. We applied group-based trajectory modeling (GBTM) to identify the patterns of HbA1C trajectories, and multiple Cox proportional hazards regressions were used to estimate the hazard ratio (HR) for the risk of HZ infection with adjustment of age, sex, and comorbidities. This study enrolled 121,999 subjects to perform the analysis. The GBTM identified four HbA1C trajectories: ‘good control’ (58.4%), ‘high decreasing’ (8.9%), ‘moderate control’ (25.1%), and ‘poor control’ (7.6%) with the mean HbA1C of 6.7% (50 mmol/mol), 7.9% (63 mmol/mol), 8.4% (68 mmol/mol), and 10.7% (93 mmol/mol) respectively. The risk of HZ was significantly higher in the poor control with an HR = 1.44 (95% CI 1.26–1.64) after adjustment for confounders and comorbidities. The risk of HZ infection for the high decreasing group (initially poor then rapidly reaching optimal control) was nonsignificant compared to the good control group. In conclusion, the patients with poor glycemic control (mean HbA1C = 10.7%) had the highest risk of HZ infection. The patients with initial hyperglycemia then reaching optimal control could have a lower risk of HZ infection.

## 1. Introduction

Herpes zoster (HZ) is a viral infection, characterized by painful, unilateral vesicular skin rash. HZ usually occurs after primary varicella infection because of reactivation of latent varicella zoster virus (VZV) which results from lower VZV-specific cell-mediated immunity [1]. HZ can contribute to several complications, such as postherpetic neuralgia (PHN) and herpes zoster ophthalmicus. Both the skin rash and complications could disturb daily activity and sleep and impair the quality of life and working performance [1]. More than 1 million individuals in the United States were diagnosed with HZ annually. It is estimated that the lifetime risk of HZ infection is approximately 30%. The incidence rate is 3 to 4 cases per 1000 person-years [1,2]. In Taiwan, the incidence rate is 3.62 to 4.89 cases per 1000 person-years in the general population and increases with older age according to the studies by using National Health Insurance Research Database (NHIRD) [3,4].

The risk of HZ infection depends on the immune status of the host, especially reduced T-cell-mediated immunity. Several immunocompromising diseases or medical conditions are the risk factors of HZ because of impaired T-cell immunity, such as malignancy, human immunodeficiency virus (HIV) disease, lymphoma, autoimmune disease, or persons receiving immunosuppressive therapy [5,6,7]. Patients with diabetes mellitus (DM) also have impaired cell-mediated cell immunity, which may result in a higher risk of infectious disease [8]. Previous studies have reported the association between HZ infection and diabetes [9,10,11,12], but the range of risk varied widely in the population of DM [13]. However, there are fewer studies to investigate the risk of HZ with uncontrolled diabetes compared to the controlled group. A nested case–control study in Israel reported the association between glycemic control and the incidence of HZ. In the population under 45 years old, the risk of HZ increased with a high level (>8% or >64 mmol/mol) of glycated hemoglobin (HbA1C) compared to the group with a low HbA1C level (<5% or <31 mmol/mol). However, this trend was not similar in older age [10].

Therefore, we used a large hospital-based cohort database in Taiwan with repeated and serial measurements of HbA1C to perform trajectory analysis in the type 2 DM patients. This study aimed to investigate the risk of HZ infection among the glycemic control that was differentiated by the distinct trajectories of HbA1C by using group-based trajectory modeling.

## 2. Materials and Methods

The Chang Gung Research Database (CGRD) is a large deidentified database in Taiwan. The database originates from the electronic medical records (EMRs) of Chang Gung Memorial Hospital (CGMH), which is the largest hospital system in Taiwan. There are seven medical hospitals, including medical centers, regional hospitals, and district hospitals, which have a total of 10,070 beds and more than 280,000 patients hospitalized annually [14]. All hospitals of this system use standardized and integrated EMRs including inpatient and outpatient visits, emergency department data, laboratory data, pathological reports, nursing records, disease category data, and operation reports [14,15]. The clinical diagnoses were identified by reading the International Classification of Diseases, 9th revision (ICD-9), and ICD-10 codes. The present study was approved by the Institutional Review Board of Chang Gung Memorial Hospital (approval number: 201801148B0D001). The Institutional Review Board waived the need for informed consent because all data were deidentified and anonymous.

### 2.1. Study Design and Population

The subjects who were diagnosed with diabetes mellitus were enrolled for the period of 1 January 2007 to 31 December 2017 (*n* = 397,993). The diabetes mellitus was confirmed by searching the ICD code (ICD-9: 250, ICD-10: E11) with at least three outpatient visits or one inpatient admission (*n* = 348,498). The date when the DM was confirmed was the index date. We extracted the laboratory data of HbA1C in the first 2 years after the index date. We excluded the subjects less than 20 years of age. Because of the possible impaired immune function, the subjects with cancer history (ICD-9: 140-208, ICD-10 C00-C96), HIV infection (ICD-9: 042, 079.53, V08; ICD-10: B20, B97.35, Z21), type 1 DM (ICD-9: 250.x1, 250.x3), systemic lupus erythematosus (SLE) (ICD-9: 710, ICD-10: M32), or rheumatoid arthritis (RA) (ICD-9: 714, ICD-10: M05, M06) were excluded. The subjects who had HZ before the index date or nearly the index date was also excluded. The live-attenuated VZV vaccine was released in 2013 in Taiwan. The vaccine was expensive in Taiwan; thus, 116 patients who had taken that vaccine were also excluded. Because the processing of the HbA1C trajectory needs at least 3 measurements of HbA1C, we excluded the subjects with less than 3 HbA1C measurements during the first 2 years. Finally, a total of 121,999 eligible subjects were enrolled for analysis. A flowchart of the study design and patients is shown in Figure 1.

### 2.2. Glycated Hemoglobin Level

The series of HbA1C within the first 2 years after the index date were extracted. In Taiwan, the recommended testing interval of HbA1C was every 3 months according to the clinical practice guideline for diabetes care published by the Diabetes Association of Taiwan [16]. The interval between each measurement should be at least 3 months; thus, we excluded the subjects with less than 3 HbA1C measurements during the first 2 years. Then, the processing of the HbA1C trajectory to identify the district glycemic pattern was performed. Besides, the last measurements of HbA1C in one year before zoster infection were collected and compared with the data in the first 2 years after the index date.

### 2.3. Outcome

The outcome of this cohort study was the new occurrence of HZ. The newly diagnosed HZ was identified by searching the ICD code (ICD-9: 053; ICD-10: B02, G53.0) which appeared the first time during the follow-up period, based on the outpatient, inpatient, and emergency department data. The subjects who had HZ before the index date of DM or HZ occurrence within 1 year after the index date were excluded. All subjects were followed until death or HZ infection until 31 December 2017.

### 2.4. Major Comorbidities

Comorbidities were identified before the index date as confounders, including hypertension (ICD-9: 40, ICD-10: I10), chronic obstructive pulmonary disease (COPD) (ICD-9: 490–491, ICD-10: J44), heart failure (ICD-9: 428, ICD-10: I50), ischemic heart disease (ICD-9: 410–414, ICD-10: I20–I25), stroke (CD-9: 433-434, ICD-10: I63, I64), and chronic kidney disease (ICD-9: 585, ICD-10: N18). The comorbidities were confirmed by at least three outpatient visits or one admission or emergency visit.

### 2.5. Statistical Analysis

We applied group-based trajectory modeling (GBTM) to identify the HbA1C trajectories. A similar developmental curve over time was recognized and classified into a subgroup [17,18]. The optimal number of subgroups should be determined. The Bayes information criterion (BIC) was used to evaluate the likelihood of the model and the number of subgroups in the model. A lower BIC value showed a better fit with the number of groups. Besides, each group needed to contain >5% population. The average posterior probabilities of group membership were calculated for each group and should be larger than 0.7.

We used the chi-square tests (categorical variables) and one-way ANOVA (continuous variables) to examine the differences among these subgroups. Multiple Cox proportional hazards regression was applied to estimate the hazard ratio (HR) for HZ infection risk of each HbA1C trajectory with adjustment of confounding factors, including age, gender, and comorbidity. The gender and age were adjusted in model 1, and comorbidities were added as confounders in model 2. Since the HbA1C trajectory pattern of the first 2 years was used to define subgroup, we tested the last year’s HbA1C before HZ infection using one-way ANOVA to compare these subgroups; once AVOVA was significant, then the post hoc test by Scheffe’s method was used to check the different HbA1C values for every other group in the four groups.

All statistical tests were two-sided and were significant if *p* < 0.05. We used SAS 9.4 (SAS Institute Inc., Cary, CA, USA) for statistical analysis and installed the PROC TRAJ application to perform the trajectory analysis [17,19,20].

## 3. Results

The trajectories of HbA1C during the first 2 years after the index date are shown in Figure 2. The series of HbA1C was processed by GBTM. The BIC became lower gradually from group 2 to group 4 with cubic order (1289537, 1246499, and 1208534, respectively). Although the addition of a fifth group has the lowest BIC (1190933), the sample size of the smallest group was below 5% (Figure 2).

Therefore, the best fit of this model was four groups with cubic orders. The largest group, with 58.4% of the patients, showed a stable pattern of good glycemic control (HbA1C 6.6–6.9% or 49–52 mmol/mol) and was named the ‘good control’ group. The second group, with 8.9% of the patients, initially showed high glycemic control during the first year but subsequently improved during follow-up (HbA1C 6.4–11.8% or 46–105 mmol/mol) and was named the ‘high decreasing’ group. The third group, with 25.1% of the patients, showed a stable pattern of moderate glycemic control (HbA1C 8.3–8.6% or 67–70 mmol/mol) and was named the ‘moderate stable’ group. The fourth group, with 7.6% of the patients, showed a stable pattern of hyperglycemia (HbA1C 10.4–11.0% or 90–97 mmol/mol) and was named the ‘poor control’ group (Figure 2).

This study enrolled 121,999 patients finally. The baseline characteristics of patients are described in Table 1. The mean age (standard deviation (SD)) was 60.4 (12.6) years and a male preponderance of 53.7%. The median (interquartile range (IQR)) follow-up period was 6.1 (3.2–9.8) years. The mean HbA1C within 2 years was 7.58% (59 mmol/mol). The average incidence of HZ from 2007 to 2017 was estimated at 3.8 cases per 1000 person-years. The patients in the 2nd to 4th groups were more likely younger and had longer follow duration compared to the good control group. The means of HbA1C within 2 years of ‘good control’, ‘high decreasing’, ‘moderate stable’, and ‘poor control’ groups were 6.7% (50 mmol/mol), 7.9% (63 mmol/mol), 8.4% (68 mmol/mol), and 10.7% (93 mmol/mol) respectively. The proportion of patients with comorbidities at baseline was higher among the patients in the good control group, compared with the other 3 groups (Table 1).

During the follow-up period (median 6.1 years, IQR 3.2–9.8 years), a total of 2867 patients suffered from HZ, about 2.4% of this population. The Kaplan–Meier curves for zoster infection by HbA1C trajectories are shown in Figure 3. The poor control groups had a higher proportion of zoster than the other groups (log-rank *p* = 0.025). The high decreasing group tended to have the lowest incidence rate.

We used multiple Cox proportional hazards regression to estimate the hazard ratio of zoster for each HbA1C trajectories, as shown in Table 2. The risk of HZ was significantly higher in the poor control group after adjusting for age and gender (Model 1) with HR 1.41 and 95% confidence interval (CI) 1.23–1.60, and it was still significantly higher after adjusting for comorbidities (Model 2) (HR 1.44, 95% CI 1.26–1.64). The other groups, the ‘moderate stable’ and the ‘high decreasing’ groups, were at higher risks, but the differences were statistically nonsignificant after adjusting for age, gender, and comorbidities. The risk of zoster was significantly higher in the poor control group than the other groups both in Model 1 and Model 2 analysis.

The means of HbA1C in the last year before HZ infection were shown in Table 3. The patients with zoster infection had a similar trend compared with the first 2 years. The average HbA1C was highest in the poor control group (mean 9.73%, SD 1.11) and lowest in the good control group (mean 6.77%, SD 0.69%). The one-way ANOVA analysis revealed a significant difference between the four groups. The post hoc tests with Scheffe’s method showed each group was significantly different from the others. In other words, the four groups’ order in the last year before HZ infection was the same as the trajectory groups in the first 2 years after the index date.

## 4. Discussion

In the present study, 58.4% of the patients maintained stable good glycemic control, and 8.9% of the patients were in the high decreasing trajectory group. Similar distinct HbA1C trajectories were observed in the other studies [18,21]. A European study from the large Diabetes Patienten Verlaufsdokumentation (DPV) multicenter diabetes registry including 6355 subjects with a 5-year follow-up period showed the largest group (56%) had persistent good control and 12% of the population showed high decreasing glycemic status in the cohort whereas 6% of the population had poor glycemic control [18]. Another study with a longer follow-up period based on Kaiser Permanente Northern California Diabetes Registry reported 82.5% of the patient had a ‘low stable’ HbA1C pattern [21]. Other trajectories, including ‘moderate increasing late’, ‘high decreasing early’, ‘moderate peaking late’, and ‘moderate peaking early’, accounted for 17.5%. This distinct pattern was different from that of previous DPV studies and our results, which may be due to the longer follow-up period (10 years). In the first 5 years, the pattern of the trajectory was like our results [21].

The patients with poor glycemic control have a higher risk of HZ infection after adjusting for confounders and comorbidities. However, the patients with initially poor control in the first year and then reaching near-optimal glycemic control subsequently had a lower risk of HZ infection and had no statistical difference from the good control group. The results suggest that the patients with poor DM control have a higher risk of HZ infection, but the risk would be lower if the hyperglycemia became controlled and maintained within optimal levels.

Several studies reported the association between DM and HZ infection [9,10,11,12]. The cell-mediated immunity (CMI) may play a key role in suppressing the reactivation of the VZV. Among the patients with uncontrolled DM, hyperglycemia compromises the CMI, including the activation of phagocytes, memory CD4+ and cytotoxic CD8+ T cells, and a variety of cytokines [22,23]. A study that was designed to investigate T-cell responses to Streptococcus pneumoniae stimulation reported that lower frequency of total CD+ T cells and diminishing memory CD4+ T-cell response were associated with elevated blood glucose and glycated hemoglobin [24]. In addition, the phagocytosis dysfunction of monocytes was found among patients with chronic hyperglycemia [25]. The individuals with short-term hyperglycemia had changes in the gene expression of cytokines and chemokines [26]. A study investigated the VZV-specific CMI which was measured by IFN-*r* ELISPOT assay among patients with DM and found that the VZV-specific CMI was lower among the patients with DM compared with healthy participants. The results suggest a higher risk of HZ infection in patients with diabetes mellitus [27]. However, few studies explored dose–effect relationships such as those between blood sugar or HbA1C levels and molecular reactions in CMI. Our finding would be a clue for further researches on increasing HbA1C to disturb the molecular reactions in CMI.

To our knowledge, the study was the first study to estimate the risk of HZ infection with a distinct pattern of glycemic control by using the trajectory model. Only a study in Israel using average HbA1C investigated the impact of glycemic control on HZ infection. That study reported that younger patients (<45 years) with worse glycemic control (HbA1C > 8%) had a higher risk of HZ infection compared with the patients with good glycemic control (HbA1C < 5%), but this finding was not seen in older age [10]. This finding was similar to ours. Another study used trajectories of HbA1C and also divided them into four groups; however, the study did not analyze specific complications of DM. Furthermore, our study used this methodology to find the risk of HZ in DM patients and similarly found two smaller groups that exhibited either persistent poor (7.6%) or initial poor (8.9%) glycemic control and then rapidly improved during the first year, which had a statistically different result of HZ infection risk. The finding led us positively to treat and encourage the patients with initially poor glycemic control.

There are some strengths of our study. First, this CGRD dataset was so large that our study enrolled 122,068 subjects to be analyzed, containing younger and elder subjects. The overall outpatient and inpatient coverage rates of the CGRD were 21.2% and 12.4% in Taiwan. The DM-specific coverage was 29% in outpatients and 16% in inpatients, almost one-third in Taiwan [14]. According to a study based on NHIRD, the average incidence of HZ infection from 2005 to 2011 was estimated at 3.6 cases per 1000 person-years [4]. This incidence was similar to our study with an incidence rate of 3.8 cases per 1000 person-years from 2007 to 2017. This is the large real database using our ordinary clinical practice. Second, our longitudinal glycemic data were analyzed in distinct trajectories. Previous studies which investigated the association between glycemic control and outcomes have used the data of HbA1C with single time or average HbA1C in the period of follow-up, considering the difference of HbA1C over time [10,28,29]. However, the character of glycemic control with HbA1C data was heterogeneous and dynamic. These approaches which used ‘‘one size fits all’’ data could not represent the actual pattern of change over time. Our trajectory method to classify the distinct glycemic pattern may fit clinical practice better and more accurately.

There were several limitations in our study. Information bias could have happened where the actual time of newly diagnosed diabetes may be before the index date in this database. Because CGMH served often as a referral hospital for advanced care, patients might be initially diagnosed or be treated somewhere else and then receive treatment in our medical system. However, DM is a chronic proceeding disease, and it is difficult to confirm the definite date of DM onset. In addition, the study was a longitudinal follow-up study using series HbA1C, and the severity of HZ infection should drive patients to visit our medical system. Consequently, the information bias did not skew our results.

The selection bias was also noted. The patients without three HbA1C measurements were excluded. These may include those who were without regular measurement but maintained good glycemic control because the medical exams may not be necessary. If this assumption is true, the results would be overestimated. However, Taiwan National Health Insurance strongly recommended checking HbA1C for DM patients, and this check was linked to insurance payment; thus, the exclusion of these patients would not likely lead to selection bias.

Third, we used the series HbA1C data for the first 2 years, not all values within the whole follow-up time. There were two reasons for this. The median (IQR) follow-up period of all subjects was 6.1 (3.2–9.8) years, and the cases with HZ infection had a shorter follow-up period (median 4.4, IQR 2.7–4.8). If we used the data for more years, the number of missing values of HbA1C would greatly increase, and the trajectory of glycemic control would not be reliable. In addition, the HZ infection could worsen glycemic control [30]. The value of HbA1C near the timing of HZ infection may be less reliable. Hence, we chose to analyze the first 2 years of HbA1C data to reduce such error. Moreover, the order of HbA1C levels among these four groups in the last year before HZ infection was similar to that in the first 2 years after the index date. This finding could prove that the HbA1C levels in each group were not changed much during the follow-up period.

## 5. Conclusions

In conclusion, the patients with poor glycemic control have a higher risk of HZ infection (HR = 1.44, 95% CI: 1.26–1.64). However, patients with initial poor glycemic control then becoming aggressively and subsequently controlled (the group of high decreasing HbA1C) would have no significant risk of HZ infection compared to the group of good control. Therefore, the management of hyperglycemia should be administered as soon as possible, and the early achievement of good glycemic control would be better to avoid HZ infection. Consequently, medical expenses could be reasonably reduced, and quality of life would be improved.

## Figures and Tables

**Figure 1 ijerph-19-02646-f001:**
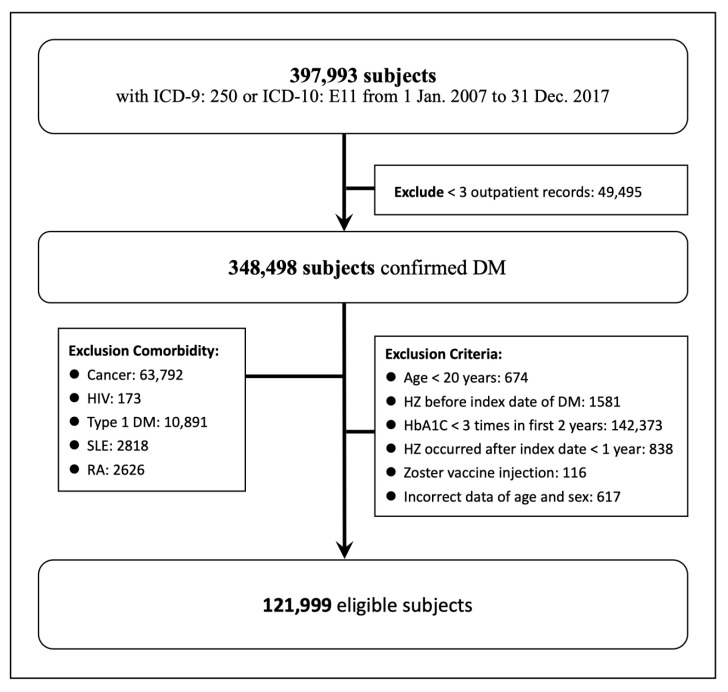
Flowchart of study design and patients.

**Figure 2 ijerph-19-02646-f002:**
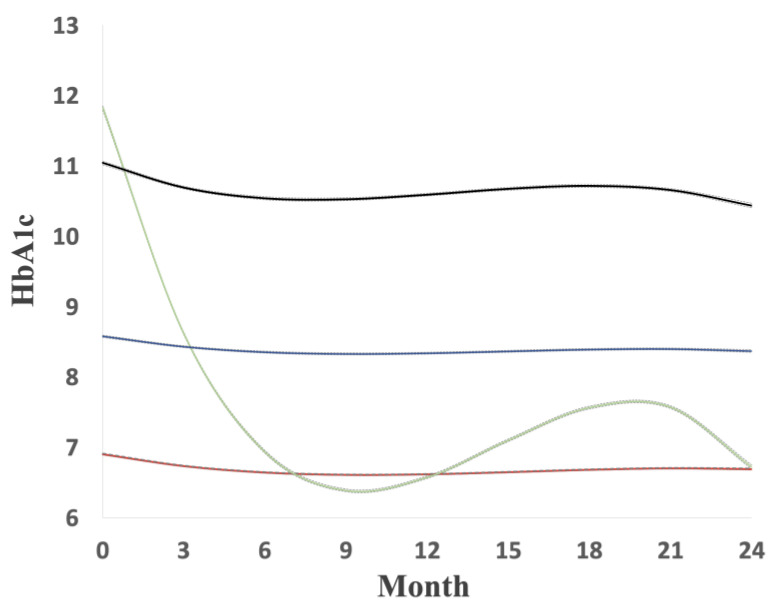
Trajectories (99% CIs) of HbA_1_C within first 2 years after index date: red line means good control group (58.4% of the population); green line means high decreasing (8.9% of the population); blue line means moderate stable (25.1% of the population); black line means poor control (7.6% of the population) (dot line means 99% confidence interval).

**Figure 3 ijerph-19-02646-f003:**
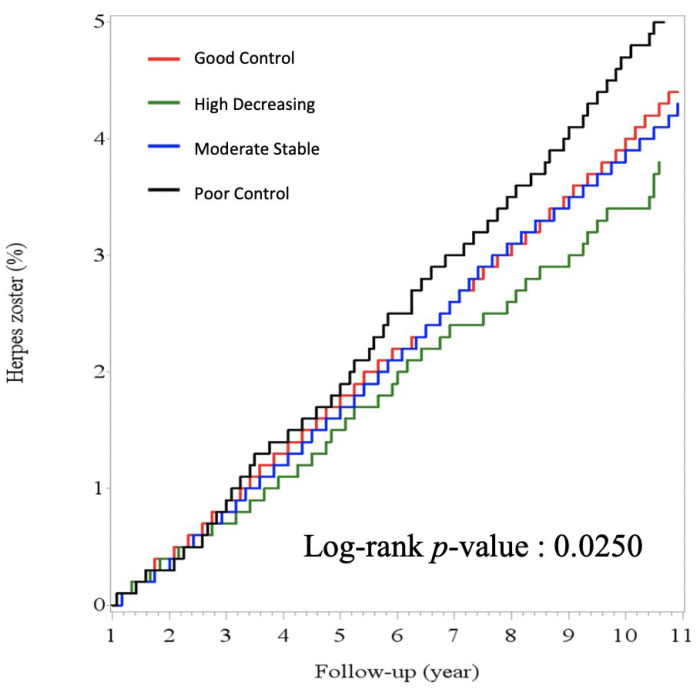
Kaplan–Meier curve for zoster infection by HbA1C trajectories. The poor control group (black line) had a higher incidence rate than the other 3 groups (log-rank *p*-value: 0.0250).

**Table 1 ijerph-19-02646-t001:** Baseline character by the groups with HbA_1_C trajectories.

	All		Group 1		Group 2		Group 3		Group 4		
			Good Control		High Decreasing		Moderate Stable		Poor Control		*p* Value
Total enrollee *n.* (%)	121,999		71,296 (58.4)		10,888 (8.9)		30,602 (25.1)		9213 (7.6)		
Herpes Zoster *n.* (%)	2867	(2.4)	1621	(2.3)	183	(1.7)	794	(2.6)	269	(2.9)	<0.0001
Age, years, mean (SD)	60.4	(12.6)	62.1	(12.4)	56.1	(12.7)	59.6	(12.1)	55.6	(12.6)	<0.0001
Sex, *n.* (%)											<0.0001
Male	65,490	(53.7)	38,046	(53.4)	6659	(61.2)	16,000	(52.3)	4785	(52.0)	
Female	56,509	(46.3)	33,250	(46.6)	4229	(38.9)	14,602	(47.7)	4428	(48.1)	
Follow date, years, median (IQR)	6.1	(3.2–9.8)	5.8	(3.0–9.4)	5.1	(2.5–8.1)	7.2	(3.8–10.8)	6.9	(3.4–10.5)	<0.0001
Mean HbA_1_C, %, mean (SD)	7.58	(1.68)	6.70	(0.79)	7.93	(2.27)	8.43	(1.17)	10.69	(1.80)	<0.0001
Comorbidity, *n.* (%)											
Hypertension	22,440	(18.4)	17,973	(25.2)	1359	(12.5)	2654	(8.7)	454	(4.9)	<0.0001
COPD	2214	(1.8)	1704	(2.4)	154	(1.4)	293	(1.0)	63	(0.7)	<0.0001
Heart failure	2701	(2.2)	2030	(2.9)	153	(1.4)	419	(1.4)	99	(1.1)	<0.0001
Ischemic heart disease	7382	(6.1)	5747	(8.1)	382	(3.5)	1052	(3.4)	201	(2.2)	<0.0001
Stroke	2958	(2.4)	2344	(3.3)	185	(1.7)	364	(1.2)	65	(0.7)	<0.0001
Chronic kidney disease	2206	(1.8)	1676	(2.4)	75	(0.7)	373	(1.2)	83	(0.9)	<0.0001

COPD, chronic obstructive pulmonary disease; IQR, interquartile range; SD, standard deviation.

**Table 2 ijerph-19-02646-t002:** Cox regression models with HbA_1_C trajectories to estimate hazard ratio of zoster.

Group	Crude HR (95% Cl)		Model 1 ^†^ HR (95% Cl)		Model 2 ^‡^ HR (95% Cl)	
HbA_1_C trajectories						
Group 1 good control	1.00	(reference)	1.00	(reference)	1.00	(reference)
Group 2 high decreasing	0.87	(0.74–1.01)	1.04	(0.89–1.21)	1.06	(0.90–1.23)
Group 3 moderate stable	0.98	(0.90–1.06)	1.06	(0.97–1.15)	1.08	(0.99–1.18)
Group 4 poor control	1.15	(1.01–1.30)	1.41	(1.23–1.60)	1.44	(1.26–1.64)
Age	1.03	(1.03–1.04)	1.03	(1.03–1.04)	1.03	(1.03–1.04)
Sex						
Female	1.00	(reference)	1.00	(reference)	1.00	(reference)
Male	0.81	(0.75–0.87)	0.91	(0.85–0.98)	0.90	(0.83–0.97)
Hypertension	1.16	(1.05–1.29)			1.01	(0.90–1.14)
COPD	1.94	(1.52–2.49)			1.63	(1.26–2.09)
Heart failure	1.18	(0.89–1.56)			0.89	(0.67–1.19)
Ischemic heart disease	1.26	(1.07–1.48)			1.10	(0.92–1.31)
Stroke	1.37	(1.09–1.74)			1.10	(0.86–1.40)
Chronic kidney disease	2.99	(2.42–3.68)			2.80	(2.26–3.46)

HR, hazard ratio; CI, confidence interval; ^†^ Model 1: adjusted for sex and age; ^‡^ Model 2: model 1 + adjusted for comorbidities.

**Table 3 ijerph-19-02646-t003:** The HbA1C in the last year before HZ infection.

	Herpes Zoster *n* (%)		HbA_1_C %, Mean (SD)		ANOVA	Post Hoc Tests by Scheffe’s Method	Difference between Means (95% CI)	
Trajectory group					*p* < 0.0001 *			
Poor control	269	(2.9)	9.73	(1.11)		Poor control vs. moderate stable	1.53	(1.38–1.68) *
						vs. high decreasing	1.99	(1.78–2.20) *
						vs. good control	2.96	(2.82–3.10) *
Moderate stable	794	(2.6)	8.20	(0.75)		Moderate stable vs. high decreasing	0.46	(0.29–0.64) *
						vs. good control	1.43	(1.34–1.52) *
High decreasing	183	(1.7)	7.74	(0.86)		High decreasing vs. good control	0.96	(0.80–1.13) *
Good control	1621	(2.3)	6.77	(0.69)				

CI, confidence interval; * *p* value < 0.05 indicates significant difference.

## Data Availability

The data could be applied to use via an application proceeding in Kaohsiung Chang Gung Memorial.

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
