# Peer review of "The Pattern of Hemoglobin A1C Trajectories and Risk of Herpes Zoster Infection: A Follow-Up Study"

_ijerph, 2022, doi:10.3390/ijerph19052646_

Round 1
Reviewer 1 Report
The questions/concerns raised in the first review of the manuscript were sufficiently adressed by the authors.
Author Response
The questions/concerns raised in the first review of the manuscript were sufficiently addressed by the authors.
Response: Thank you very much.
Reviewer 2 Report
The topic is interesting, actually, the manuscript speaks about the risk of herpes in poor glycemic control diabetes patients.
The manuscript is well written, the methodology clearly reported, and the results clearly presented.
Line 210 needs references.
In conclusions, please add some public health considerations of your results
Reviewer 3 Report
Since patient with diabetes mellitus (DM) have impaired cell-mediated cell immunity, which may result in higher risk of infection disease, the authors have investigated the risk of herpes Zoster HZ infection in 121,999 DM patients. The major outcome is that patients with the poor glycemic control have the higher risk of HZ infection after adjusting the confounders and comorbidities. Patients with initially poor control in the first year and improved glycemic control subsequently had lower risk of HZ infection and had no statistically difference from the good control group.
The results suggest that the patients with poor DM control have higher risks of HZ infection, but the risk becomes lower if the hyperglycemia becomes controlled and maintained within optimal levels.
Several typos and some English needs to be corrected.
After all, the question arises, whether HZ affect DM?
No further comments.
Author Response
Please see the attachment.

This manuscript is a resubmission of an earlier submission. The following is a list of the peer review reports and author responses from that submission.
Round 1
Reviewer 1 Report
The authors should be congratuladed for their large observational study adressing an inportant question, if glycemic control in DM is responsable for the already known increased risk for HZ.
major comments:
My major concern with the design of the study ist, that authors postulate that the glycemic control in the first 2 years is responsible for HZ some years later. This is counterintuitive to previous data, showing that current glycemic control is linked to current infection risk in general.
If autors chose to define glycemic control in the first two years, they should at least report last HbA1c before HZ as an additional variable and test this varriable as a confounder.
minor comments:
Line 89: Rheumatoid Arthritis is defined by more ICD10 codes: M05 and M06
Reviewer 2 Report
Pan et al. present a study titled "The Pattern of Hemoglobin A1C Trajectories and Risk of Herpes Zoster Infection: a Follow-up Study". The database they provided is sufficient however study design and data analysis have significant flaws.
- The trajectory of HbA1C was for 2 years however the HZ development was monitored for up to 11 years. It does not provide any association between such a short trajectory to long-term disease development.
- The labeling of four groups such as "good control" or "high stable' were not consistent in the manuscript therefore it is impossible to comprehend.
- Figures and tables were not cleared explained with a detailed figure legend and it is difficult to understand.